# Abnormalities on Perfusion CT and Intervention for Intracranial Hypertension in Severe Traumatic Brain Injury

**DOI:** 10.3390/jcm9062000

**Published:** 2020-06-25

**Authors:** Shannon Cooper, Cino Bendinelli, Andrew Bivard, Mark Parsons, Zsolt J. Balogh

**Affiliations:** 1Department of Traumatology, John Hunter Hospital Newcastle, Newcastle, NSW 2305, Australia; shannon.cooper@health.nsw.gov.au (S.C.); cino.bendinelli@health.nsw.gov.au (C.B.); 2School of Medicine and Public Health, University of Newcastle, Newcastle, NSW 2300, Australia; andrew.bivard@newcastle.edu.au (A.B.); mark.parson@health.nsw.gov.au (M.P.); 3Department of Neurology, University of Melbourne, Melbourne, VIC 3050, Australia

**Keywords:** perfusion CT, traumatic brain injury, intracranial hypertension, intracranial pressure monitoring

## Abstract

The role of invasive intracranial pressure (ICP) monitoring in patients with severe traumatic brain injury (STBI) remain unclear. Perfusion computed tomography (CTP) provides crucial information about the cerebral perfusion status in these patients. We hypothesised that CTP abnormalities would be associated with the severity of intracranial hypertension (ICH). To investigate this hypothesis, twenty-eight patients with STBI and ICP monitors were investigated with CTP within 48 h from admission. Treating teams were blind to these results. Patients were divided into five groups based on increasing intervention required to control ICH and were compared. Group I required no intervention above routine sedation, group II required a single first tier intervention, group III required multiple different first-tier interventions, group IV required second-tier medical therapy and group V required second-tier surgical therapy. Analysis of the results showed demographics and injury severity did not differ among groups. In group I no patients showed CTP abnormality, while patients in all other groups had abnormal CTP (*p* = 0.003). Severe ischaemia observed on CTP was associated with increasing intervention for ICH. This study, although limited by small sample size, suggests that CTP abnormalities are associated with the need to intervene for ICH. Larger scale assessment of our results is warranted to potentially avoid unnecessary invasive procedures in head injury patients.

## 1. Introduction

In patients with severe traumatic brain injury (STBI), interventions are aimed at maintaining cerebral perfusion pressure (CPP) by minimising intracranial hypertension (ICH). Intracranial pressure (ICP) monitoring can be used to guide intervention, however, these monitors are invasive and associated with the risk of complications such as haemorrhage and intracranial infection. Recent studies have questioned the benefit of ICP monitoring [1,2], and, as a result, there is currently no consensus on patient selection for ICP monitor insertion [3]. Neuroimaging is used to guide decisions about which patients may benefit from an ICP monitor. Although non-contrast CT (NCCT) provides anatomical information, it does not provide direct information about cerebral perfusion. Cerebral perfusion CT (CTP) is a logistically non-demanding technique that does provide prompt information on cerebral perfusion [4]. CTP involves continual scanning while a bolus of intravenous contrast transits through the brain vasculature. It has a well-established role in the management of strokes and guides decisions on thrombolytic therapy by defining the area of ischaemic penumbra [5,6,7,8]. In studies in STBI patients, ICH has been associated with lower cerebral blood flow on CTP [9]. Abnormalities on CTP have also been shown to correlate with abnormalities in CPP and loss of cerebral autoregulation [10,11,12,13]. Several small studies have used CTP to investigate the cerebral blood flow changes after intervention for ICH in humans [14,15,16], and in animal models [17], CTP has been shown to predict short- and long-term functional outcomes in both mild and severe traumatic brain injury [18,19,20,21,22,23,24,25,26].

No studies have yet examined the ability of CTP to predict the need for intervention for ICH. The current study sought to investigate whether abnormalities or ischaemia detected on CTP were associated with intervention for ICH. We hypothesised that abnormalities or ischaemia on CTP would be associated with more intervention to lower ICP as well as a higher likelihood of needing second tier interventions to control ICH refractory to first-tier interventions.

## 2. Materials and Methods

This study was undertaken at the John Hunter Hospital, a Level 1 trauma centre and obtained ethics approval from the Hunter New England Ethics Committee (09/12/16/5.01). A retrospective analysis of all STBI patients with a functioning ICP monitor who were investigated with CTP within 48 h from admission between 2009 and 2014 was undertaken. CTP was performed in eligible patients that the treating neurosurgical team judged as requiring a progress NCCT within 48 h of admission. This was usually performed due to clinical deterioration or failure to progress. All ICP monitors were inserted after the initial CT and 12 to 48 h prior to CTP. The inclusion criteria were: age above 18 years of age and Glasgow Coma Score (GCS) of less than nine before intubation. Exclusion criteria were: pregnancy, intravenous contrast allergy, renal impairment and haemodynamic instability. All CTP images were reviewed and reported by consensus by two consultant stroke-neurologists and scored on two binary variables. First, whether there was any perfusion abnormality on CTP or not. Second, whether there were any perfusion abnormalities in the severely ischaemic range or not (defined as a delay in mean transient time of more than two seconds compared to the normal reference artery). The CTP was performed for research purposes and this analysis did not occur in real-time. Thus, the results were not available to the treating team to impact treatment decisions.

All patients had ICP and intervention data collected from intensive care unit (ICU) records. The type of ICP monitor, whether external ventricular drain (EVD) or Codman strain gauge was recorded. According to institutional protocols, all patients were sedated with a midazolam and/or propofol infusion for the duration of ICP monitoring. Episodes of ICH were defined as an ICP greater than 20 mmHg and interventions were initiated in a stepwise manner. Similarly to previous studies and guidelines and reflecting the clinical practice at our institution, interventions were divided into first tier and second tier interventions [27,28]. First tier interventions included the use of extra boluses of sedating agents, paralysing agents, hyperosmolar agents or cerebrospinal fluid (CSF) drainage for those patients with an EVD. Second tier therapies used were either barbiturate infusion or surgical decompressive craniectomy. Other therapies such as cooling or hyperventilation were difficult to ascertain consistently from ICU records and so were not recorded. Patients were divided into 5 groups of increasing intervention for ICH. Group I had no episodes of ICH and required no intervention above routine sedation. Group II had episodes of ICH that were responsive to a single additional first-tier therapy. Group III required multiple first-tier therapies to manage ICH (i.e., hypertonic saline and CSF drainage). Group IV required second-tier medical therapy. Group V required second-tier surgical therapy.

Two post hoc analyses were performed. In the first, we compared patients who required interventions for ICH (groups II, III, IV and V combined) versus those who did not (group I). In the second post hoc analysis we compared patients who required a second-tier intervention for ICH (groups IV and V, combined) and those who did not (groups I, II and III).

Additional data collected were age, gender and mechanism of injury, GCS prior to intubation, injury severity score (ISS) and head and neck abbreviated injury score (HNAIS), lactate and base deficit on arrival. NCCT findings were categorised using the Rotterdam CT classification [29]. In-hospital outcome data included mortality and length of stay in ICU. Long term functional outcome data was collected at 6 months by a rehabilitation physician using the Glasgow Outcome Scale Extended (GOSE) [30] which was then dichotomised into favourable outcome (GOSE 5–8) and unfavourable outcome (GOSE 1–4).

All statistical analyses were carried out using IBM SPSS version 24 (Armonk, New York, United States). Normality for all continuous data was tested using the Shapiro-Wilk test. Normally distributed continuous data were presented as a mean and standard deviation. When compared, student’s *t*-test was used, or ANOVA if there were more than 2 groups. Continuous data that did not conform to a normal distribution were presented with a median and interquartile range (IQR) and compared using a Wilcoxon sign-rank test for paired observations, a Mann–Whitney U test for unpaired groups, or a Kruskal–Wallis test for comparisons with more than two groups. Dichotomous and categorical data were presented as a percentage of the total number of relevant patients in that category. These data were compared using a Chi-square test or Fisher’s exact test if any cell’s expected value was less than 5. For all comparisons, a *p* value of less than 0.05 was considered significant.

## 3. Results

Figure 1 shows the screening process and the number of patients who received an ICP monitor and were investigated with a CTP.

The demographic details of the study population are included in Table 1. These reflect the typical demographics of STBI: patients were young (median age 33), predominantly male (82%), with median pre-intubation GCS of 5, median HNAIS of 5 and were severely injured (mean ISS 31). The median Rotterdam NCCT score was 2. Systemic hypoperfusion was common on arrival (median lactate 3; median base deficit 3). Sixteen (57%) of the 28 patients had an EVD inserted while the rest received a Codman strain gauge. Mean ICU length of stay was 13 days (SD 5.4) and 5 patients (18%) died. Outcomes were generally unfavourable, with only nine (32%) patients with a favourable outcome at six months follow up.

Table 2 illustrates the groups of patients stratified by the ICH interventions provided. Two (7%) patients had no episodes of ICH and did not require any intervention other than routine sedation (group I). Five (18%) patients required single first tier therapy (Group II). Eleven (39%) required multiple first tier therapies (Group III). Five (18%) required second tier medical therapy (Group IV) and five (18%) more patients required second tier surgical therapy (Group V). The statistically significant differences observed between the groups were: Rotterdam score (4 versus 2; *p* = 0.046) and ICU days (8 vs. 18; *p* = 0.03). In group I none had abnormality detected on CTP, while patients in all other groups showed abnormal CTP (*p* = 0.003). Severe ischaemia observed on CTP was associated with increasing intervention for ICH. Severe ischaemia was observed in 0%, 25%, 36%, 40% and 60% in group I, group II, group III, group III, group IV and group V, respectively. These differences failed to reach statistical significance (*p* = 0.091). The percentage of patients with a CTP abnormality and the percentage of patients with severe ischaemia in each group are summarized graphically in Figure 2.

In Table 3 patients were dichotomised into two groups, those who did not require any intervention for ICH versus those who required an intervention for ICH. Two (7%) patients did not require any intervention for ICH and 26 (93%) patients required some form of intervention for ICH. There were no statistically significant differences between the groups on demographic, clinical or outcome variables. None of the patients who did not require intervention had any CTP abnormality, while all patients who required an intervention had an abnormality on CTP. This was the only difference that reached statistical significance (*p* = 0.003). Ten (38%) of the patients who needed an intervention for ICH had severe ischaemia on CTP compared with neither of the patients who did not need intervention, but this difference failed to reach significance.

In Table 4 patients were dichotomised into two groups: those who did not require second tier intervention versus those who required second-tier intervention. Eighteen (64%) patients did not require any second tier intervention and 10 (36%) required second-tier intervention. The only statistically significant difference between the two groups was ICU stay (11 versus 17 days; *p* = 0.007). Sixteen (89%) patients in the non-second-tier group and 10 (100%) patients in the second-tier group had an abnormal CTP, however this difference did not reach statistical significance. Five (29%) patients in the non-second tier group showed severe ischaemia compared with 5 (50%) patients in the second-tier group (*p* = 0.415).

## 4. Discussion

Despite over a decade of research on CTP in the STBI population, there is still a paucity of studies with relatively small sample sizes. Several studies have investigated the potential uses for CTP in quantifying cerebral perfusion abnormalities in STBI [10,11,12,13], as well as changes after treatment [14,15,16,17] and prognostication [18,19,20,21,22,23,24,25,26]. This information may be helpful to guide the selection of patients for invasive ICP monitoring. Predicting which patients will require intervention to control ICH and therefore may benefit from ICP monitoring is challenging. Previous Brain Trauma Foundation guidelines have recommended ICP monitoring in all patients with an STBI and an abnormal NCCT [27]. However, these recommendations were based on descriptive studies, and the most recent Brain Trauma Foundation Guidelines concluded there was insufficient evidence to guide physicians in selecting patients for ICP monitoring [3]. Given the current uncertainty around the indications for invasive ICP monitoring in STBI patients, we have investigated if CTP could be used to predict which patients are likely to need intervention for ICH and would therefore require an ICP monitoring device. In our cohort of STBI patients, the only two patients who required no intervention above routine sedation were also the only two patients who had no abnormality detected on CTP, a difference that was statistically significant. In addition, although the comparison failed to reach statistical significance, there did appear to be a signal towards increased likelihood of severe ischaemia on CTP and intensity of treatment for ICH. It appears that CTP could play a role in predicting the role of ICP monitors and ICH occurrence, however, our results must be interpreted with great caution due to the small sample size. To address this shortcoming, a follow up analysis with groups dichotomised based on whether they required only first-tier interventions or required second-tier interventions likewise demonstrated a signal in the expected direction but did not reach statistical significance. It is highly likely that the current study was insufficiently powered to detect any difference between the groups.

One of the strengths of the current study is that it is the largest series of STBI patients examined by whole brain CTP to date. The sample size compares favourably with other studies examining CTP in TBI patients using CTP [10,11,12,13,14,15,16,17,18,19,20,21,22,23,24,25,26]. Furthermore, we selected and investigated a quite homogenous population of extremely injured STBI patients (CTP was performed only when, despite 24–48 h of ICU management, neurologic status was either not improving or deteriorating).

This study has several limitations that need to be expressed. We acknowledge that the ICP data was limited by the ICU records. There was considerable variety in the method of recording the hourly ICP as well as any ICP spikes and the administration of any treatment. The recording of timed ICP data was not consistent enough to allow meaningful examination of the ICP at the time of CTP. The interventions that were used for this study were selected because generally these were the most reliably recorded. Other interventions such as cooling or hyperventilation were not recorded consistently enough to enable meaningful use. Some patients had multiple repeated uses of different first-tier therapies to control ICH, while others had thiopentone, a second-tier therapy, after failure to respond to relatively fewer first tier therapies. This could have resulted in greater heterogeneity in the groups, and a prospective study capturing ICP and intervention data with the clinical context of the patient may be able to divide patients into more homogenous ICP intervention groups.

## 5. Conclusions

In conclusion this study did suggest that intervention for ICH above routine sedation was associated with an abnormal CTP, while severe ischaemia on CTP was associated with increasing levels of intervention for ICH. Although the size of the study does not allow firm recommendations, it is likely that early CTP in STBI patients may help in deciding which patients could avoid the invasiveness of ICP monitors.

## Figures and Tables

**Figure 1 jcm-09-02000-f001:**
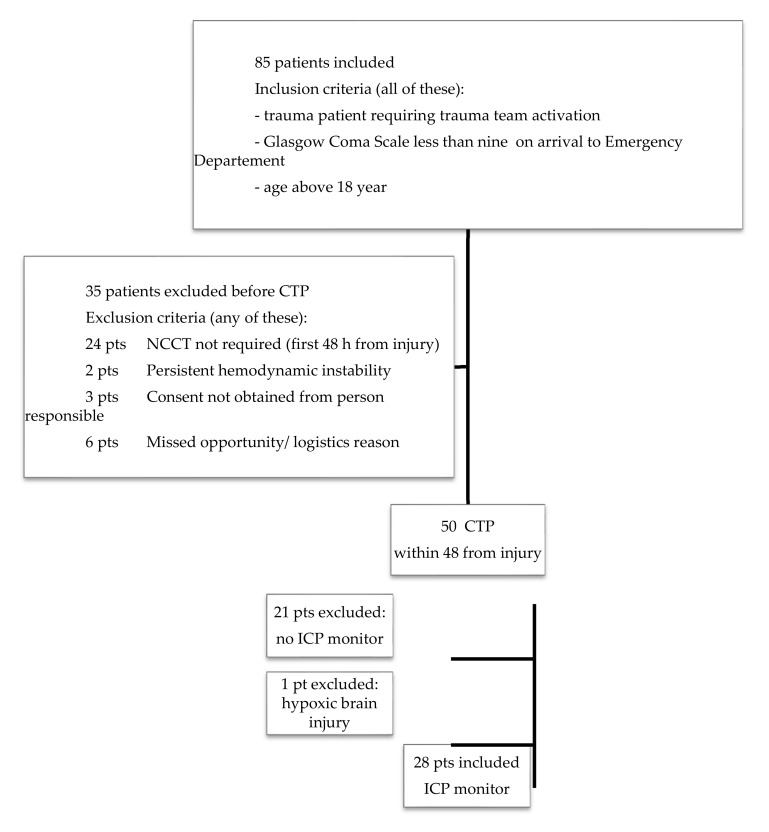
Eligible patients, excluded patients and included patients. CTP = brain perfusion computed tomography; NCCT = non-contrast CT; ICP = intracranial pressure.

**Figure 2 jcm-09-02000-f002:**
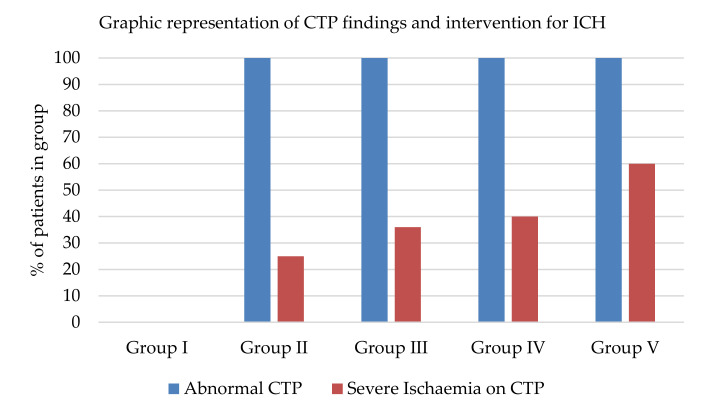
Graphic representation of CTP findings and intervention for ICH. CTP = brain perfusion computed tomography; ICH = intracranial hypertension.

**Table 1 jcm-09-02000-t001:** Demographic, clinical and outcome details.

Number of Patients	28
Age (Years): Median (IQR)	33 (22–53)
Male: *n* (%)	23 (82%)
Clinical Variables	
EVD: *n* (%)	16 (57%)
Pre-Intubation GCS: Median (IQR)	5 (3–7)
ISS: Mean (SD)	31 (8.2)
HNAIS: Median (IQR)	5 (4–5)
Lactate: Median (IQR)	3 (2–4)
BD: Median (IQR)	3 (2–5)
Rotterdam: Median (IQR)	2 (2–3)
Outcome Variables	
ICU Days: Mean (SD)	13 (5.4)
Mortality: *n* (%)	5 (18%)
Favourable GOSE at 6 Months: *n* (%)	9 (32%)

BD = base deficit on arrival expressed in mEq/L; CTP = brain perfusion computed tomography; EVD = external ventricular drain; GCS = Glasgow coma scale; GOSE = Glasgow outcome scale extended; HNAIS = head and neck abbreviated injury score; ICP = intracranial pressure; ICU = intensive care unit; IQR = interquartile range; ISS = injury severity score; Lactate = lactate on arrival expressed in mmol/L; NCCT = non-contrast brain CT; SD = standard deviation.

**Table 2 jcm-09-02000-t002:** Demographic, clinical, outcome and CTP findings of patients stratified into groups by increasing level of intervention for ICH.

	Group I	Group II	Group III	Group IV	Group V	*p*-Value
Number of Patients: *n* (%)	2 (7%)	5 (18%)	11 (39%)	5 (18%)	5 (18%)	
Age: Median (IQR)	23(19–23)	35 (23–57)	33 (20–48)	27 (23–57)	46 (29–60)	N.S. ^b^
Male: *n* (%)	2 (100%)	4 (80%)	8 (73%)	4 (80%)	5 (100%)	N.S. ^a^
Clinical Variables						
EVD: *n* (%)	0 (0%)	3 (60%)	5 (46%)	4 (80%)	4 (80%)	N.S. ^a^
GCS: Median (IQR)	5 (3–5)	4 (4–7)	5 (3–6)	5 (3–7)	5 (3–9)	N.S. ^a^
ISS: Mean (SD)	27 (3.5)	34 (8.8)	31 (9.8)	33 (4.8)	27 (7.8)	N.S. ^c^
HNAIS: Median (IQR)	5 (4–5)	4 (4–5)	5 (4–5)	5 (4–5)	5 (3–6)	N.S. ^b^
Lactate: Median (IQR)	3 (2–3)	3 (2–7)	3(2–4)	3 (2–4)	3 (2–4)	N.S. ^b^
BD: Median (IQR)	5 (5–6)	2 (4–6)	4 (2–7)	3 (1–3)	4 (1–2)	N.S. ^b^
Rotterdam: Median (IQR)	2 (1–2)	2 (2–3)	2 (2–3)	2 (2–2)	4 (3–4)	0.046 ^b^
Outcome Variables						
ICU Days: Mean (SD)	8 (2.1)	14 (4.8)	10 (4.9)	16 (5.9)	18 (1.8)	0.03 ^c^
Mortality: *n* (%)	0 (0%)	1 (20%)	2 (18%)	1 (20%)	1 (20%)	N.S. ^a^
Favourable GOSE: *n* (%)	2 (100%)	1 (20%)	3 (27%)	2 (40%)	1 (20%)	N.S. ^b^
CTP Results						
Abnormal: *n* (%)	0 (0%)	5 (100%)	11 (100%)	5 (100%)	5 (100%)	0.003 ^a^
Severe Ischaemia: *n* (%)	0 (0%)	1 (25%)	4 (36%)	2 (40%)	3 (60%)	N.S. ^a^

^a^ = Fisher’s exact test; ^b^ = Kruskal–Wallis test, ^c^= one-way ANOVA test. BD = base deficit on arrival expressed in mEq/L; CTP = brain perfusion computed tomography; EVD = external ventricular drain; GCS = Glasgow coma scale; GOSE = Glasgow outcome scale extended; HNAIS = head and neck abbreviated injury score; ICP = intracranial pressure; ICH = intracranial hypertension; ICU = intensive care unit; IQR = interquartile range; ISS = injury severity score; Lactate = lactate on arrival expressed in mmol/L; NCCT = non-contrast brain CT; SD = standard deviation; N.S. = not significant (*p* > 0.05).

**Table 3 jcm-09-02000-t003:** Demographic, clinical, outcome and CTP findings dichotomised into groups by need for any intervention for ICH.

	No Intervention	Intervention	*p*-Value
Number of Patients: *n* (%)	2 (7%)	26 (93%)	
Age: Median (IQR)	23 (NA)	33 (18–49)	N.S. ^c^
Male: *n* (%)	2 (100%)	21 (81%)	N.S. ^a^
Clinical Variables			
EVD: *n* (%)	0 (0%)	16 (62%)	N.S. ^a^
Pre-Intubation GCS: Median (IQR)	5 (NA)	5 (3–7)	N.S. ^a^
ISS: Mean (SD)	27 (3.5)	31 (8.1)	N.S. ^c^
HNAIS: Median (IQR)	4.5 (NA)	5 (4–6)	N.S. ^b^
Lactate: Median (IQR)	2.6 (NA)	2.8 (2.0–3.6)	N.S. ^b^
BD: Median (IQR)	5 (NA)	2.3 (0.6–4.2)	N.S. ^b^
Rotterdam: Median (IQR)	2 (NA)	2 (1–3)	N.S. ^b^
Outcome Variables			
ICU Days: Mean (SD)	8 (2.1)	13 (5.3)	N.S. ^c^
Mortality: *n* (%)	0 (0%)	5 (19%)	N.S. ^a^
Favourable Outcome: *n* (%)	2 (100%)	7 (27%)	N.S. ^a^
CTP Findings			
Abnormal: *n* (%)	0 (0%)	26 (100%)	0.003. ^a^
Severe Ischaemia: *n* (%)	0 (0%)	10 (38%)	N.S. ^a^

^a^ = Fisher’s exact test; ^b^ = Mann–Whitney U test, ^c^ = student’s *t*-test. BD = base deficit on arrival expressed in mEq/L; CTP = brain perfusion computed tomography; EVD = external ventricular drain; GCS = Glasgow coma scale; GOSE = Glasgow outcome scale extended; HNAIS = head and neck abbreviated injury score; ICP = intracranial pressure; ICH = intracranial hypertension; ICU = intensive care unit; IQR = interquartile range; ISS = injury severity score; Lactate = lactate on arrival expressed in mmol/L; NCCT = non-contrast brain CT; SD = standard deviation; N.S. = not significant (*p* > 0.05).

**Table 4 jcm-09-02000-t004:** Demographic, clinical, outcome and CTP findings dichotomised into groups by need for second-tier interventions.

	No Second-Tier Intervention	Second-Tier Intervention	*p*-Value
Number of Patients: *n* (%)	18 (64%)	10 (36%)	
Age: Median (IQR)	30 (22–49)	45 (25–57)	N.S. ^b^
Male: *n* (%)	14 (78%)	9 (90%)	N.S. ^a^
Clinical Variables			
EVD: *n* (%)	8 (44%)	8 (80%)	N.S. ^a^
Pre-Intubation GCS: Median (IQR)	5 (3–6)	5 (3–7)	N.S. ^a^
ISS: Mean (SD)	31 (9.0)	30 (6.8)	N.S. ^c^
HNAIS: Median (IQR)	5 (4–5)	5 (4–5)	N.S. ^b^
Lactate: Median (IQR)	3 (2–4)	2.9 (2–4)	N.S. ^b^
BD: Median (IQR)	4 (1–6)	3 (1–3)	N.S. ^b^
Rotterdam: Median (IQR)	2 (1–2)	2 (2–2)	N.S. ^b^
Outcome Variables			
ICU Days: Mean (SD)	11 (4.9)	17 (4.5)	0.007 ^c^
Mortality: *n* (%)	3 (17%)	2 (20%)	N.S. ^a^
Favourable Outcome: *n* (%)	6 (64%)	3 (30%)	N.S. ^a^
CTP Findings			
Abnormal: *n* (%)	16 (89%)	10 (100%)	N.S. ^a^
Severe Ischaemia: *n* (%)	5 (29%)	5 (50%)	N.S. ^a^

^a^ = Fisher’s exact test; ^b^ = Mann–Whitney U test, ^c^ = student’s *t*-test. BD = base deficit on arrival expressed in mEq/L; CTP = brain perfusion computed tomography; EVD = external ventricular drain; GCS = Glasgow coma scale; GOSE = Glasgow outcome scale extended; HNAIS = head and neck abbreviated injury score; ICP = intracranial pressure; ICU = intensive care unit; IQR = interquartile range; ISS = injury severity score; Lactate = lactate on arrival expressed in mmol/L; NCCT = non-contrast brain CT; SD = standard deviation; N.S. = not significant (*p* > 0.05).

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
