# Peer review of "Abnormalities on Perfusion CT and Intervention for Intracranial Hypertension in Severe Traumatic Brain Injury"

_jcm, 2020, doi:10.3390/jcm9062000_

Round 1

Reviewer 1 Report

The authors report a retrospective analysis of patients with severe traumatic brain injury with intracranial pressure monitoring who also had perfusion CT imaging. They attempt to correlate perfusion abnormalities with need for tiered ICP interventions. They found that the two patients who required no intervention had no perfusion deficits while the 26 patients with tiered interventions had perfusion abnormalities. Because these results had a p value of 0.003, the authors conclude that normal perfusion CT is not associated with need for ICP intervention. One must remember that the population studied is very small. If the next patient without intervention had an abnormal perfusion CT, then the results would not likely be regarded as statistically significant. Therefore, the conclusion is stated too strongly for the the quality of the data. A more accurate conclusion would be that patients who require tiered ICP intervention are likely to have perfusion CT abnormalities.

Other changes needed to improve the quality of the manuscript includes the following:

  1. The authors need to define when the perfusion CT was performed. Was it performed prior to the first intervention?  Or was it performed after interventions were started? This timing should be discussed in the context of the perfusion CT abnormality.
  2. The authors should provide the indications for perfusion CT. How many severe traumatic brain injury patients did not have perfusion CT and so were excluded? Why did this 28 get perfusion CT?
  3. The authors do have some ICP data which they did not include. What was the ICP and CPP closest to the perfusion CT?

Author Response

Thankyou very much for your suggestions.

The study is limited by the small sample size, and particularly by the fact that only 2 patients had no abnormality on CTP as mentioned in the paper. The conclusion has been changed to the less overstated conclusion that abnormality on CTP scans was associated with intervention for ICH above routine sedation.

1) All ICP monitors were inserted after initial admission NCCT, 12-48 hours before CTP. All CTP scans were performed at the time of progress NCCT, within the first 48 hours of admission. This information has been added to the paper

2) The indication for perfusion CT were patients who were judged by the treating neurosurgical team to require a progress NCCT within the first 48 hours of admission - usually for deterioration or failure to progress. This information has been added to the paper. A flowchart has also been included to indicate the number of severely head injured patients who were identified as eligible for inclusion - but did not proceed to progress NCCT, as well at the number of patients who had a CTP, but did not have an ICP monitor and so weren't included in the study.

3) The ICP and CPP at time of perfusion CT was inconsistently recorded in the ICU notes with variable methods of recording the data. This data was not used because it was unreliable with considerable variation in how close to the time of perfusion CT the last ICP was recorded.

Reviewer 2 Report

The authors are to be commended. They report a well thought out initial study attempting to assess a relationship between CTP imaging abnormalities and the severity of intervention needed to control intracranial hypertension (ICH) among patients with severe traumatic brain injuries (sTBI). The authors realistically identify and address the limitations of their study, and frame their results and conclusions appropriately. I agree with the authors' assessment that a larger study of similar nature is warranted and needed.

The only major addition to this manuscript that I would recommend is a graphical representation of the primary outcome that succinctly represents the major takeaway from this manuscript. If possible, the authors might consider plotting each case on a simple graph with increasing interventions on the Y-axis, CTP grading on the X-axis, and a line of correlation with appropriate statistical summary (i.e. r^2).

Overall, this is a quality manuscript. Congratulations are due to the authors.

Author Response

Thankyou very much for your comments and great suggestion on how to graphically make clear the key take home point of the paper.

Because the CTP data is binary (perfusion abnormality y/n and severe ischaemia y/n) the scatterplot of all the cases by group did not meaningfully convey the data. In order to graphically present the key point of the data, a chart has been constructed plotting the percentage of patients with CTP abnormality in each group as well as the percentage of patients in each group with evidence of severe ischaemia on CTP. This illustrates that as the groups require more intervention to control ICH, they also have a greater proportion of patients who have severe ischaemia on CTP

Round 2

Reviewer 1 Report

The authors have satisfactorily addressed the reviewers comments.

Author Response

.